# SECURING DEEP GENERATIVE MODELS WITH ADVERSARIAL SIGNATURE

## ABSTRACT

Recent advances in deep generative models have led to the development of methods capable of synthesizing high-quality, realistic images. These models pose threats to society due to their potential misuse. Prior research attempted to mitigate these threats by detecting generated images, but the varying traces left by different generative models make it challenging to create a universal detector capable of generalizing to new, unseen generative models. In this paper, we propose to inject a universal adversarial signature into an arbitrary pre-trained generative model, in order to make its generated contents more detectable and traceable. First, the imperceptible optimal signature for each image can be found by a signature injector through adversarial training. Subsequently, the signature can be incorporated into an arbitrary generator by fine-tuning it with the images processed by the signature injector. In this way, the detector corresponding to the signature can be reused for any fine-tuned generator for tracking the generator identity. The proposed method is validated on the FFHQ and ImageNet datasets with various state-of-the-art generative models, consistently showing a promising detection rate. Code will be made publicly available.

## 1 INTRODUCTION

Recent advances in deep generative models Pan et al. (2019); Yang et al. (2022) have enabled the generation of highly realistic synthetic images, which benefits a wide range of applications such as neural rendering Isola et al. (2017); Park et al. (2019); Chan et al. (2022); Li et al. (2020), text-to-image generation Ramesh et al. (2022); Reed et al. (2016); Zhang et al. (2017); Saharia et al. (2022), image inpainting Pathak et al. (2016); Yu et al. (2018), super-resolution Ledig et al. (2017), among others. As a side effect, synthetic but photo-realistic images pose significant societal threats due to their potential misuse, including copyright infringement, the dissemination of misinformation, and the compromise of biometric security systems.

To mitigate these risks, one straightforward approach is to imprint digital watermarks on generated images during a post-processing phase. However, this post-processing step is usually decoupled from the main model, making it difficult to enforce. Therefore, recent work has focused on a more enforceable solution: using a deep model as a detector to identify synthetic images Nirkin et al. (2021); Liu et al. (2021b); Guarnera et al. (2020); Wang et al. (2022); Sha et al. (2022). They manifest effectiveness against known generators, *i.e.*, those involved in the training dataset, but suffer from a performance drop against unseen generators. This is due to the variability of the model "signatures" hidden in the generated images across different models, as illustrated in Fig. 1 (a). Consequently, these detection-based systems require frequent retraining upon the release of each new generator to maintain their effectiveness, which is impractical in real-world applications.

In this work, we propose a more robust approach to identify synthetic images by integrating a model-agnostic "signature" into any pre-trained generator. Since the signature is concealed within the model parameters, it becomes non-trivial for malicious users to erase, and is inevitably included in the generated contents, thereby facilitating detection. By using a universal signature (*i.e.*, model-agnostic signature), we can leverage the same detector to identify images from different generators, eliminating the need for retraining the detector with the introduction of new generators.

To determine the optimal signature for images, we first train a signature injector $W$ in an adversarial manner against a classifier $F$ (the detector). In particular, the injector $W$ learns to add a minimal

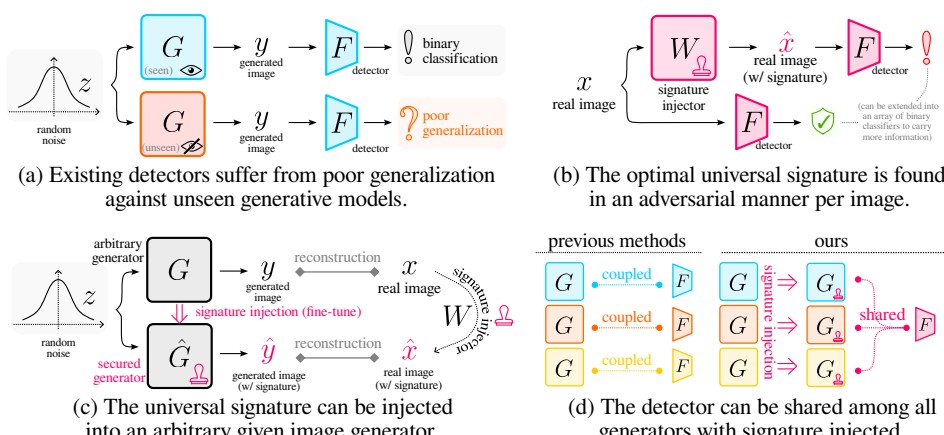

Figure 1: Illustration of securing deep generative models through universal adversarial signature.

alternation $\kappa$ to a given image $\boldsymbol{x}$ to produce a slightly modified image $\hat{\boldsymbol{x}}$. The injector aims to make the alternation $\kappa$ as small as possible in order to retain image quality, while simultaneously maximizing the detector's accuracy, as shown in Fig. 1 (b). Importantly, the detector $F$ is not necessarily designed to be a binary classifier. It can be a multi-class classifier that produces a multi-bit binary code to convey additional information to help track and identify the source of a generated image.

To implant such signatures into an arbitrary pre-trained image generative model $G$, we fine-tune $G$ using a set of images processed by $W$, resulting in a secured generator $\hat{G}$, as demonstrated in Fig. 1 (c). Images generated by the secured generator $\hat{G}$ can be identified by $F$ as $\hat{G}$ inherits the signatures from $W$. In addition, as shown in Fig. 1 (d), the detector $F$ is no longer associated with specific generators during the training process, and therefore can be shared across different generators. As the injector $W$ and detector $F$ can be reused for different pre-trained generators, the adversarially learned signature becomes universal (model-agnostic) among all secured generators.

To demonstrate the effectiveness of the proposed method, we conduct extensive experiments on the FFHQ Karras et al. (2019) and ImageNet Deng et al. (2009) datasets. Three state-of-the-art generative models, namely LDM Rombach et al. (2022), ADM Dhariwal & Nichol (2021), and StyleGAN2 Karras et al. (2020), are used in evaluations. The experimental results demonstrate that a given generative model can learn to add the adversarial signatures to its outputs, making them more recognizable by the generated image classifier, while not sacrificing the image quality.

The contributions of this paper are summarized as follows:

1. We propose to learn the optimal universal signatures through adversarial learning against a classifier (*i.e.* the detector).
2. We propose to inject the universal adversarial signatures to secure an arbitrary pre-trained image generative model. Secured generative models can share the same detector.
3. Our proposed universal adversarial signature is capable of carrying additional information, such as the generator identity, for tracking the source of the images.

## 2  RELATED WORKS

**Deep Generative Models** Pan et al. (2019); Yang et al. (2022) have been greatly improved recently, enabling realistic large-scale image and text synthesis Ramesh et al. (2022), Brown et al. (2020). This field has undergone various mainstream models, including autoregressive models Salimans et al. (2017), variational autoencoders Huang et al. (2018), normalizing flows Kingma & Dhariwal (2018), generative adversarial models (GANs) Goodfellow et al. (2014); Karras et al. (2020); Esser et al. (2021), and more recently, denoising diffusion models (DDMs) Ho et al. (2020); Rombach et al. (2022). In particular, GANs and DDMs are capable of imposing threats to society due to their potential abuse. This paper focuses on mitigating their potential threats.

**Generated Image Detection** is committed to mitigating the potential threats of the generated images. The existing methods extract features to discover artifacts either in the spatial domain Nirkin et al.

(2021); Guarnera et al. (2020); Sha et al. (2022) or frequency domain Liu et al. (2021b); Wang et al. (2022). However, these passive detection models may not generalize well for unseen generative models. In this paper, learning the adversarial signature entails actively modifying a generator to make its outputs more recognizable, which is different from the existing work focused on generated image detection. The scope of this paper is general-purpose generated image detection, which is not limited to a specific type of media such as deepfake.

**Image Watermarking** Cox et al. (2007); Begum & Uddin (2020) can limit the misuse of digital images, including the generated ones. Although watermarks vary in their visibility Nikolaidis & Pitas (1998); Zhou et al. (2018), it is difficult for them to achieve robustness, imperceptibility, and high capacity at the same time Begum & Uddin (2020). Besides, deep-learning-based methods involve adding templates to the real images Asnani et al. (2022), or inserting watermarks into the generated image Zhao et al. (2023a). However, these methods are subject to an impractical assumption that malicious users will apply watermarks. Instead, we modify generative models to make adversarial signatures inevitable.

**Neural Network Fingerprinting** addresses the challenges of visual forensics and intellectual property protection posed by deep models Peng et al. (2022); Liu et al. (2021a). Model-specific GAN fingerprints, either learned Yu et al. (2019) or manually-crafted Marra et al. (2019), can be used for generated image detection and source tracking, but still has to be re-trained against new generators. In contrast, our detector can be reused for any future generator.

## 3   OUR APPROACH

Given a set of images $X$, a typical deep generative model $G$ learns the underlying distribution $p_X$ of $X$, and can generate a realistic image from a random noise $z$, $i.e.$, $y \triangleq G(z)$. Due to the threats posed by the potential abuse of the outputs from the generator $G$, it is necessary to develop a classifier $F$ to distinguish the generated (signed) images from real ones, where $F(\cdot) \in (0, 1)$. A real image $x \in X$ is labeled as $0$, while a generated image $y$ is labeled as $1$.

As discussed in Section 1, we explore modifying the parameters of a given generator $G$, to make its outputs more recognizable by $F$, and hence securing the generator $G$. Our approach is a two-stage method. Firstly, a signature injector $W$ learns to generate adversarial signatures, while a classifier $F$ learns to detect them. The signature injector $W$ is subsequently used for teaching an arbitrary generative model $G$ to generate recognizable images. The proposed method is illustrated in Figure 1 and summarized in Algorithms 1-2.

### 3.1   OPTIMAL ADVERSARIAL SIGNATURE

Consider a system consisting of a signature injector $W$ and a classifier $F$. In the optimal case, $F$ can discriminate the signed images from clean images based on the subtle and imperceptible alternation made by $W$ (imperceptibility). The system is robust to image restoration attack if augmented by noise (persistency), $i.e.$ the signature cannot be removed by an image restoration model $M$. The following propositions state the imperceptibility and persistency of the adversarial signatures in detail.

**Proposition 3.1.** *(**Imperceptibility**) There exist optimal pairs of signature injector $W$ and classifier $F \colon \mathbb{R}^n \mapsto \{0, 1\}$, so that for any image $\forall x \in \mathbb{R}^n$, $\forall \epsilon > 0$, its distance to the signed image $W(x)$ is smaller than $\epsilon$, and $F$ correctly discriminates them,* i.e., $\|W(x) - x\| < \epsilon$, and $F(W(x)) \neq F(x)$.

**Proposition 3.2.** *Let $e$ be a zero-mean, positive-variance additive noise. There exist noise augmented $W, F$ that satisfy the following condition: $\forall \epsilon > 0, \mathbb{E}_e[\|W(x + e) - x\|] < \epsilon$ and $F(W(x)) \neq F(x)$.*

**Proposition 3.3.** *(**Persistency**) The noise augmented $W, F$ stated in Proposition A.3 is robust to image restoration attack, as optimizing $\min_M \mathbb{E}_{x,e}[\|M(W(x + e)) - x\|]$ will result in $M$ being an identity mapping.*

*Proof.* Please refer to the appendix.

*Remark* 3.4. Intuitively, when $W(x + e)$ is close enough to $x$, the training of $M$ to remove signatures tends to fall into a trivial sub-optimal solution of copying the input to the output. Therefore, even if $W$ is disclosed to malicious users, it is still difficult to erase the signature.

| **Algorithm 1:** Training Signature Injector | **Algorithm 2:** Securing an Image Generator |
|---|---|
| 1: **Input:** A set of images $X$; | 1: **Input:** A set of clean images $X$; |
| 2: **Output:** (1) Signature injector $W$; | 2:        A pre-trained generator $G$; |
| 3:        (2) Binary classifier $F$; | 3:        The signature injector $W$; |
| 4: Randomly initialize $W$ and $F$; | 4: **Output:** A fine-tuned generator $\hat{G}$; |
| 5: **for** $i = 1$ **to** MaxIteration_stage1 **do** | 5: $\hat{G} \leftarrow G$; |
| 6:     Randomly sample $\boldsymbol{x} \in X$; | 6: $\hat{X} \leftarrow \{W(\boldsymbol{x}) \| x \in X\}$; |
| 7:     $\hat{\boldsymbol{x}} \leftarrow W(\boldsymbol{x})$; | 7: **for** $i = 1$ **to** MaxIteration_stage2 **do** |
| 8:     $L_{\text{rec}} \leftarrow \|\hat{\boldsymbol{x}} - \boldsymbol{x}\|_2^2$; | 8:     Randomly sample a noise $\boldsymbol{z}$; |
| 9:     Random transformation for $\boldsymbol{x}$ and $\hat{\boldsymbol{x}}$; | 9:     Randomly sample $\hat{\boldsymbol{x}} \in \hat{X}$; |
| 10:    $L_{\text{cls}} \leftarrow \log F(\boldsymbol{x}) + \log(1 - F(\hat{\boldsymbol{x}}))$; | 10:    Update $\hat{G}$ using $\hat{G}(\boldsymbol{z})$ and $\hat{\boldsymbol{x}}$; |
| 11:    $L \leftarrow L_{\text{rec}} + \lambda \cdot L_{\text{cls}}$; | 11: **end for** |
| 12:    $\Delta W, \Delta F \leftarrow \partial L / \partial W, \partial L / \partial F$; | |
| 13:    $W, F \leftarrow \text{Adam}(W, F, \Delta W, \Delta F)$; | |
| 14: **end for** | |

## 3.2 Universal Adversarial Signature Injector

Given an image $\boldsymbol{x}$, the signature injector model $W$ adds an imperceptible alternation to it, resulting in a "signed" image $\hat{\boldsymbol{x}} \triangleq W(\boldsymbol{x})$, of the same size as $\boldsymbol{x}$. The difference $\kappa \triangleq \hat{\boldsymbol{x}} - \boldsymbol{x}$ is termed as the "adversarial signature", which varies with the input image $\boldsymbol{x}$ and the injector $W$. Meanwhile, the classifier $F$ aims to discriminate the signed image $\hat{\boldsymbol{x}}$ from the clean image $\boldsymbol{x}$. In this paper, the signed image $\hat{\boldsymbol{x}}$ is labeled as $1$, while the clean image $\boldsymbol{x}$ is labeled as $0$.

To find the desired pair of $W$ and $F$ as discussed above, the goal is to ensure that the signed image $\hat{\boldsymbol{x}}$ is as close to the clean image $\boldsymbol{x}$ as possible, while the classifier $F$ should correctly recognize the signed images. The goal can be expressed as the following optimization problem:

$$\min_{W,F} \mathbb{E}_{\boldsymbol{x}} \|W(\boldsymbol{x}) - \boldsymbol{x}\|_2^2, \quad \text{s.t.} \quad \mathbb{E}_{\boldsymbol{x}} \left[ F(\boldsymbol{x}) + (1 - F(W(\boldsymbol{x}))) \right] = 0. \tag{1}$$

By introducing the Lagrange multiplier, we obtain the following loss function:

$$\mathcal{L} = \mathbb{E}_{\boldsymbol{x}} [\underbrace{\|W(\boldsymbol{x}) - \boldsymbol{x}\|_2^2}_{L_{\text{rec}}} + \lambda \underbrace{(F(\boldsymbol{x}) + 1 - F(W(\boldsymbol{x})))}_{L_{\text{cls}}}]. \tag{2}$$

The $L_{\text{rec}}$ term in Eq. (2) is the mean squared error that enforces the signatures to be imperceptible (not obviously impacting the image quality). The $L_{\text{cls}}$ term can be seen as a classification loss that encourages the classifier to distinguish the signed images from the clean images.

In practice, we find directly optimizing Eq. (2) through gradient descent methods results in $\lambda = 0$, and the model copying the input to the output. Therefore, we empirically fix $\lambda$ to a small value. In addition, we replace the $L_{\text{cls}}$ part with the commonly used cross-entropy loss. Therefore, $W$ and $F$ are jointly trained by optimizing the following approximated loss function:

$$L = \mathbb{E}_{\boldsymbol{x} \sim p_X} \{L_{\text{rec}}(\boldsymbol{x}; W) + \lambda \cdot L_{\text{cls}}(\boldsymbol{x}; W, F)\}, \tag{3}$$

$$\text{where} \quad L_{\text{rec}}(\boldsymbol{x}; W) = \|W(\boldsymbol{x}) - \boldsymbol{x}\|_2^2, \tag{4}$$

$$\text{and} \quad L_{\text{cls}}(\boldsymbol{x}; W, F) = \log F(\boldsymbol{x}) + \log(1 - F(W(\boldsymbol{x}))). \tag{5}$$

During training, the signature injector $W$ and the generated image classifier $F$ are, in fact, adversarial against each other. The minimization of $L_{\text{cls}}$ requires the injector $W$ to add a sufficiently large and easy-to-identify signature $\kappa$ to make $\hat{\boldsymbol{x}}$ separatable from $\boldsymbol{x}$; while the minimization of $L_{\text{rec}}$ requires the signature injector $W$ to shrink the norm of $\kappa$ for the sake of its imperceptibility, which makes the signed image $\hat{\boldsymbol{x}}$ more difficult to be separated from $\boldsymbol{x}$.

The overall process of this stage is summarized in Algorithm 1. Note, to make the signature $\kappa$ robust, both the original image $\boldsymbol{x}$ and the signed image $\hat{\boldsymbol{x}}$ are transformed before being fed to $F$. The transformations involve commonly used augmentation operations, which are detailed in Section 4.

Albeit our method slightly resembles letting $W$ produce adversarial examples to flip the prediction of $F$, it is totally different from adversarial attack. Compared to, *e.g.*, C&W attack Carlini & Wagner (2017), our method generates the signed images in a single forward pass (instead of iteratively), and jointly trains $F$ (instead of freezing its parameters), which is totally different from adversarial attack.

**Binary Code Extension.** By extending the binary classifier $F$ to multiple outputs, the adversarial signature will be able to carry additional information such as generator identity for tracking the source of the generated images. To achieve this, we can first determine the binary code length as $n$ bits, which directly decides the number of all possible binary codes as $2^n$. The selection of $n$ ($n > 0$) depends on the number of user-predefined messages to be represented by the binary codes. For instance, when $n = 2$, the binary codes for generators are `01`, `10`, and `11`, while the code `00` is reserved for real images. During the training process, a random binary code except for `00` from the $2^n - 1$ possible binary codes is chosen for every generated image. Next, the single classification layer in $F$ is extended into $n$ classification layers in parallel for binary code prediction.

Meanwhile, the binary code is converted by a code embedding module into an embedding vector. It comprises two linear layers and SiLU Hendrycks & Gimpel (2016) activation. The binary code embedding is fed into injector $W$ via AdaIN Karras et al. (2019) after every convolution layer for modulating the signatures. Note, when $n = 1$ (default), a constant vector is used as the embedding.

### 3.3 Securing Arbitrary Generative Model

In order to make the adversarial signatures inevitable, it would be better if they could be integrated into the model parameters through, for example, fine-tuning. In this way, the outputs from the generators will be detectable by $F$, and hence the generative model is secured. Therefore, in this stage, the signature injector $W$ will process the training data, based on which an arbitrary given (pre-trained) generative model is fine-tuned to learn the adversarial signatures. This conceptually shifts the distribution the generator has learned towards the distribution of the signed images.

Specifically, given a set of training images $X$, the already trained signature injector $W$ is used to apply an adversarial signature to each image $\boldsymbol{x} \in X$, resulting in a signed image $\hat{\boldsymbol{x}}$. Assume we have an arbitrary already trained deep generative model $G$, which can generate an image $\boldsymbol{y}$ from a random noise $\boldsymbol{z}$, *i.e.*, $\boldsymbol{y} = G(\boldsymbol{z})$. Then, the model $G$ is fine-tuned using the signed images, resulting in the model $\hat{G}$, which generates a signed image $\hat{\boldsymbol{y}}$ from a random noise $\boldsymbol{z}$, *i.e.*, $\hat{\boldsymbol{y}} = \hat{G}(\boldsymbol{z})$. By default, the concrete loss function during fine-tuning is consistent with the original training loss of $G$. An optional loss term, *i.e.*, $\xi \cdot \log(1 - F(\hat{G}(\boldsymbol{z})))$ can be appended to guide the training of $\hat{G}$ using the trained classifier $F$ (fixed), where $\xi$ is a constant that controls the weight of this loss term. The overall procedure of stage two is summarized in Algorithm 2.

As the fine-tuning process is agnostic to generator architecture, it is applicable to a wide range of generative models, including but not limited to GANs Pan et al. (2019) and DDMs Yang et al. (2022). As the $W$ and $F$ are fixed in the second stage, they are reusable for different generators.

**Binary Code Extension.** In this stage, a binary code can be assigned to a specific $G$. Every signed image $\hat{\boldsymbol{x}}$ for fine-tuning $G$ is generated by $W$ with the assigned code.

**Inference Stage.** As the fine-tuned model $\hat{G}$ is expected to learn the signatures, the classifier $F$ from the first stage can be directly used to identify whether $\hat{\boldsymbol{y}}$ is a generated (signed) image.

## 4 Experiments

In this section, we demonstrate the effectiveness of the proposed method through experiments. Our method is implemented in PyTorch Paszke et al. (2019). The code will be released in the future.

**Datasets & Models.** We adopt the U-Net Ho et al. (2020) architecture for signature injector $W$, and ResNet-34 He et al. (2016) as the classifier $F$. The proposed method is evaluated with two datasets: FFHQ Karras et al. (2019) and ImageNet Deng et al. (2009); using three generative models: LDM Rombach et al. (2022), ADM Dhariwal & Nichol (2021), and StyleGAN2 Karras et al. (2020) at $256 \times 256$ resolution. We use their official training code for the experiments, except for StyleGAN2. A third-party implementation[1] is used for StyleGAN2. We sample 1,000 images from FFHQ as the test set and use the remaining images for training. For experiments on ImageNet, we use the official training split for training, and sample 1,000 images from the validation split as our test set. The image quality (FID, PSNR) and classification accuracy (Acc, ROC) are evaluated on the test sets (1,000

---

[1] `https://github.com/rosinality/stylegan2-pytorch`

Table 1: Validating $W$ and $F$ in the first stage when the length of the binary code is $n = 1$. The symbols "↑" and "↓" mean "the higher the better" and "the lower the better", respectively.

| Dataset | Signature Injector $W$ | | Classifier $F$ |
| --- | --- | --- | --- |
| | PSNR ↑ | FID ↓ | Accuracy (%) ↑ |
| FFHQ | 51.4 | 0.52 | 100.0 |
| ImageNet | 38.4 | 5.71 | 99.9 |

Table 2: Validating $W$ and $F$ in the first stage when the length of the binary code is $n = 2$. The symbols "↑" and "↓" denote "the higher the better" and "the lower the better", respectively.

| Dataset | Signature Injector $W$ | | Classifier $F$ |
| --- | --- | --- | --- |
| | PSNR ↑ | FID ↓ | Accuracy (%) ↑ |
| FFHQ | 44.9 | 2.68 | 99.9 |

Figure 3: Sample outputs from the signature injector $W$ in stage one. The two columns on the left correspond to FFHQ, while the rest correspond to ImageNet. The signature $\kappa$ is visualized as the pixel-wise L-2 norm, where the peak value varies across inputs.

images). The only exceptions are the FID scores in Tab. 3, which are evaluated on 50K randomly generated images against the corresponding training sets following Rombach et al. (2022).

**Hyper-Parameters.** In stage one, the balance factor $\lambda$ in Eq. (3) is set as $0.05$ for the FFHQ dataset, and $1.0$ for the ImageNet dataset. The batch size is set as $24$. The models are trained using the Adam Kingma & Ba (2014) optimizer for $10^6$ iterations, with the learning rate of $10^{-4}$. In stage two, we follow the parameter settings of the respective generative models. The parameter $\xi$ is empirically set as $0.05$ for StyleGAN2, and $0$ for the remaining models.

**Data Augmentation.** The image transformation operations used to process $x$ and $\hat{x}$ for training $F$ are random rotation (the angle is uniformly sampled within $[-30°, 30°]$), random horizontal flip (with $0.5$ probability), and Gaussian blur (the variance is uniformly sampled within $[0.01, 10]$). Any output of $W$ and input to $F$ will be clipped to $[0, 1]$, and padded with the smallest constant error to make it an integer multiple of $1/255$, to ensure validity as an image.

**Binary Code.** By default, the binary code length is $n = 1$, which means $F$ only predicts whether the input is generated or not. For the $n > 1$ case, we specifically choose $n = 2$ to ensure a certain level of generator diversity, while avoiding some unnecessary experiment cost for demonstration.

**Evaluation Protocol.** The experimental results are reported based on the test sets. (1) Signature injector $W$: The $\kappa$ is expected to be imperceptible to retain the image quality of $\hat{x}$ compared to $x$. Therefore, $W$ is quantitatively evaluated using the peak signal-to-noise ratio (PSNR) and FID Heusel et al. (2017) of its outputs. (2) Generated image classifier $F$: The generated/real binary classification and binary code prediction are evaluated in classification accuracy. (3) Generator $\hat{G}$: the fine-tuning process of $\hat{G}$ is expected to make $\hat{G}$ add adversarial signatures while retaining image quality. Hence, the FID of the generated signed image $\hat{y}$ and the accuracy of $F$ against $\hat{G}$'s outputs is reported.

## 4.1 VALIDATING $W$ AND $F$ IN THE FIRST STAGE

In the first stage, we validate the signature injector $W$ for the image quality, and the classifier $F$ for the accuracy against the outputs of $W$. The experiments are conducted on FFHQ and ImageNet, respectively. The corresponding results are shown in Table 1 and Table 2 for the $n = 1$ case and the $n = 2$ case, respectively.

According to Table 1, when the binary code is 1-bit, the adversarial signature can be added to

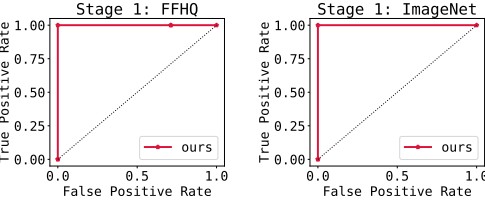

Figure 2: The ROC curves of $F$ against $W$'s outputs on the FFHQ (left) and ImageNet (right) datasets in the first stage ($n = 1$).

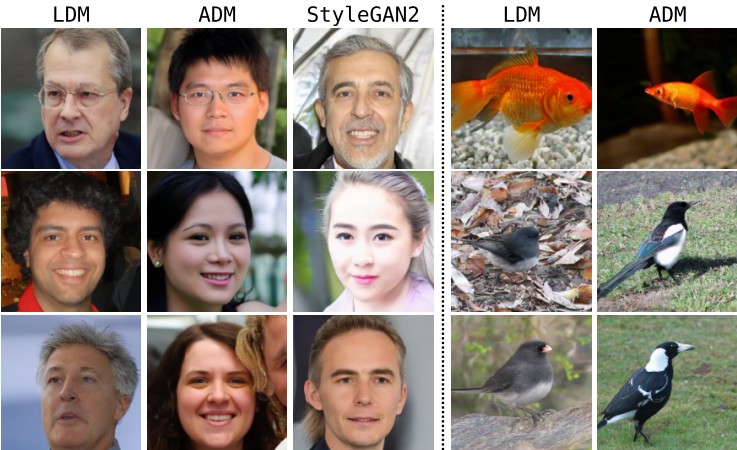

LDM  ADM  StyleGAN2  LDM  ADM

Figure 4: Sample outputs from the fine-tuned generator $\hat{G}$ in stage two. The three columns on the left correspond to the FFHQ dataset, while the two on the right correspond to the ImageNet dataset.

Table 3: Validating $\hat{G}$ and $F$ in the second stage when the length of binary code is $n = 1$. "FID*" is for the official pre-trained model; "FID" is for the model reproduced with the respective official code; "Acc." is generated/real classification accuracy.

Table 4: Validating $\hat{G}$ and $F$ in the second stage when the length of binary code is $n = 2$. See the caption for Table 3 for the meaning of "FID*", "FID", and "Acc.".

| Dataset | Generator | $G$ FID* ↓ | $G$ FID ↓ | $\hat{G}$ FID ↓ | $F$ Acc. (%) ↑ |
|---|---|---|---|---|---|
| FFHQ | LDM | 9.36 | 9.70 | 9.20 | 100.0 |
| | ADM | - | 12.32 | 13.61 | 100.0 |
| | StyleGAN2 | 4.16 | 3.97 | 4.35 | 99.9 |
| ImageNet | LDM | 7.41 | 6.72 | 5.48 | 100.0 |
| | ADM | 6.38 | 7.36 | 6.65 | 100.0 |

| Dataset | Model (code) | $G$ FID* ↓ | $G$ FID ↓ | $\hat{G}$ FID ↓ | $F$ Acc. (%) ↑ |
|---|---|---|---|---|---|
| FFHQ | LDM (01) | 9.36 | 9.70 | 9.86 | 99.8 |
| | LDM (10) | | | 9.20 | |
| | LDM (11) | | | 10.35 | |

the outputs of $W$ while retaining good image quality. This is reflected by $51.4$ PSNR and $0.52$ FID on FFHQ, and $38.4$ PSNR and $5.71$ FID on ImageNet. The results on ImageNet (natural images) are slightly worse than that on FFHQ (face images) due to the more complex distribution. Some images with signatures are visualized in Figure 3.

Apart from the injector, the classifier $F$ achieves $100.0\%$ and $99.9\%$ accuracy on FFHQ and ImageNet, respectively. The corresponding ROC curves can be found in Figure 2. These results suggest that although the learned signatures are small in L-2 norm, they are still highly recognizable by $F$.

According to Table 2, when the binary code length is $n = 2$, our method remains effective, as suggested by the good image quality for $W$ and high classification accuracy of $F$. Notably, since the $n = 2$ case requires $W$ to learn different variants of $\kappa$ for different binary codes, the learning becomes more difficult than in the $n = 1$ case, resulting in a slight performance gap.

## 4.2 VALIDATING $\hat{G}$ AND $F$ IN THE SECOND STAGE

In the second stage, a pre-trained $G$ is fine-tuned with $W$ and $F$ being fixed. We conduct experiments accordingly to validate the fine-tuned generator $\hat{G}$, and the classifier $F$ against the outputs of $\hat{G}$. The results can be found in Table 3 and Table 4 for the $n=1$ and $n=2$ cases, respectively.

According to Table 3, when the binary code length is $n = 1$, the generator $\hat{G}$ can learn the adversarial signatures from $W$, which makes its outputs more recognizable by $F$. Take the LDM model on the FFHQ dataset as an example. The fine-tuned model $\hat{G}$ achieves a similar FID to its original counterpart $G$. This indicates no significant output quality difference between $G$ and $\hat{G}$. To demonstrate this qualitatively, we visualize some generated images in Figure 4.

Although the adversarial signatures the generator $\hat{G}$ has "inherited" are imperceptible, they are still highly recognizable by $F$. This is quantitatively demonstrated by the $100.0\%$ generated/real image classification accuracy. The corresponding ROC curves can be found in Figure 5.

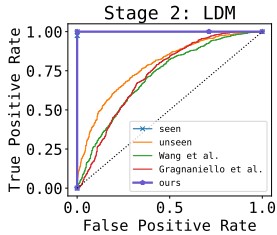 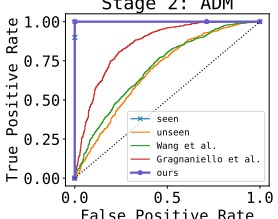 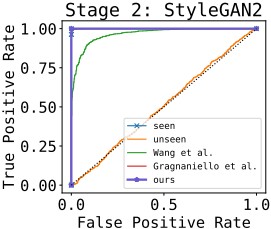

Figure 5: The ROC curves of $F$ against $\hat{G}$ (for our method) and $G$ (for baseline method) in the second stage ($n = 1$). The baseline method involves the "seen" case and the "unseen" case.

Table 6: Model sensitivity to $\lambda$ on FFHQ dataset.

|  |  | 1.0 | 0.1 | 0.05 | 0.01 | 0.005 | 0.001 | 0.00001 | 0.0 |
|---|---|---|---|---|---|---|---|---|---|
| | $\lambda$ | | | | | | | | |
| $W$ | PSNR ↑ | 37.5 | 42.8 | 51.4 | 52.1 | 52.7 | 54.9 | 63.0 | 67.5 |
| | FID ↓ | 9.33 | 3.16 | 0.52 | 0.46 | 0.44 | 0.39 | 0.06 | 0.02 |
| $F$ | Acc. (%) ↑ | 100.0 | 100.0 | 100.0 | 100.0 | 100.0 | 100.0 | 99.6 | 50.0 |

Table 7: Robustness of adversarial signature after common image transformations.

| Transformation | Gaussian blur | Horizontal flip | Rotation |
|---|---|---|---|
| $F(\hat{G}(\boldsymbol{z}))$ Acc. (%) ↑ | 100.0 | 100.0 | 100.0 |

Table 8: Validating $W$ and $F$ when they are separately trained. The length of the binary code is $n = 1$. The dataset is FFHQ.

| Model | Signature Injector $W$ | | Classifier $F$ |
|---|---|---|---|
| | PSNR ↑ | FID ↓ | Accuracy (%) ↑ |
| Jointly | 51.4 | 0.52 | 100 |
| Separately | 23.4 | 140 | 50.1 |

Table 9: Validating $W$, $F$, and $M$ in the first stage with 1-bit binary code. The noise is Gaussian with zero mean and $0.5$ standard deviation.

| Setting | $W$ | $F(W(\boldsymbol{x}))$ | $F(M(W(\boldsymbol{x})))$ |
|---|---|---|---|
| | PSNR ↑ | Acc. (%) ↑ | Acc. (%) ↑ |
| (w/o noise $\boldsymbol{e}$, w/o $L_{\text{aux}}$) | 51.4 | 100.0 | 50.0 |
| (w/o noise $\boldsymbol{e}$, w/ $L_{\text{aux}}$) | 44.8 | 99.9 | 52.6 |
| (w/ noise $\boldsymbol{e}$, w/o $L_{\text{aux}}$) | 29.4 | 99.9 | 50.0 |
| (w/ noise $\boldsymbol{e}$, w/ $L_{\text{aux}}$) | 29.1 | 99.9 | 98.2 |

According to Table 4, when the binary code length is $n = 2$, the adversarial signatures can also be effectively learned by the generators, which can still be detected by $F$.

## 4.3 COMPARISON TO STATE-OF-THE-ART METHODS

After verifying the effectiveness of our proposed method, we compare it with a baseline method and the state-of-the-art methods on FFHQ.

The baseline method corresponds to directly training the classifier $F$ (ResNet-34) to differentiate the generated images $\boldsymbol{y}$ from the original images $\boldsymbol{x}$. As shown in the first row of Table 5, if all three generators (*i.e.*, LDM, ADM, and StyleGAN2) are *seen* by $F$, its accuracy is close to $100\%$. However, in the second row, the baseline method suf-

Table 5: Generated image detection accuracy with 95% error bars. The first four rows are based on the official pre-trained generators. The last row is based on fine-tuned generators.

| Detector \ Generator | LDM | ADM | StyleGAN2 |
|---|---|---|---|
| Baseline (Seen) | 99.9±.006 | 99.7±.003 | 99.9±.006 |
| Baseline (Unseen) | 51.6±.031 | 49.8±.031 | 49.9±.031 |
| Wang et al. (2020) | 50.4±.031 | 49.9±.031 | 66.7±.029 |
| Gragnaniello et al. (2021) | 50.2±.031 | 49.9±.031 | 97.8±.009 |
| Ours | 100.0±.00 | 100.0±.00 | 99.9±.006 |

fers from poor generalization against *unseen* generators under the leave-one-out setting. For instance, in the first column, the ADM and StyleGAN2 are seen by $F$, but not LDM. The accuracy of $F$ against the LDM outputs drops to mere $51.6\%$. The corresponding ROC curves can be found in Figure 5.

The generalization issue against *unseen* generators also exists with the state-of-the-art methods including Wang et al. (2020); Gragnaniello et al. (2021), as shown in Table 5. In contrast, our method can reuse the $W$ and $F$ for any generative model, and achieve high accuracy as long as its input is from a fine-tuned generator.

## 5 DISCUSSIONS

In this section, we study the sensitivity of $\lambda$ in Eq.(3), and some alternative method designs. We also discuss how the desired characteristics mentioned in Section 1 are satisfied.

## 5.1 PARAMETER SENSITIVITY OF $\lambda$ & PRE-TRAINED $F$

**Sensitivity of $\lambda$.** In Eq. (3), the parameter $\lambda$ balances the two loss terms $L_{\text{rec}}$ and $L_{\text{cls}}$, which are adversarial against each other as discussed in Section 3.2. We conduct experiments with varying $\lambda$ values on FFHQ for the first stage, in order to study the sensitivity of $\lambda$. The results are shown in

Table 6. When $\lambda$ is gradually decreased from $1.0$, the accuracy of $F$ is not very sensitive. However, a clear trend can be seen where $W$ tries to sacrifice image quality in exchange for a lower cross-entropy loss. When $\lambda = 0$, $W$ is expected to learn the identity mapping, and $F$ is not trained. As a result, the reconstructed image is of high quality, and $F$ behaves the same as a random classifier. Most importantly, a nearly optimal pair of $W$ and $F$ can be found even if $\lambda$ is very small, which leads to a negligible image quality drop. This supports our theory in Proposition A.1.

**Pre-trained $F$.** To better understand the distinction between adversarial signatures and the features used by baseline detectors, we replace the $F$ with the pre-trained and fixed "Baseline (Seen)" classifier from Table 5 in the first stage. This leads to significantly worse performance as shown in Table 8. The results suggest that there is hardly any resemblance in features between our signature-based classifier and a baseline classifier. Therefore, the adversarial signature is different from the features used by the baseline detectors, and $W$ and $F$ should be jointly optimized.

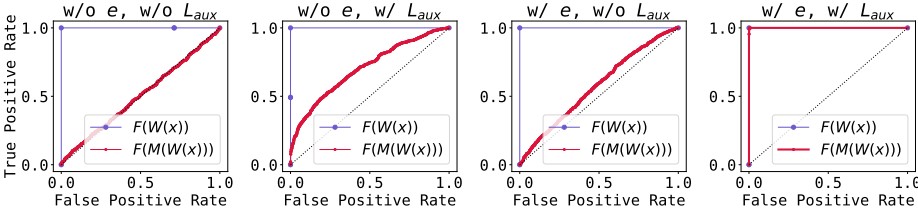

Figure 6: ROC for $F(W(\boldsymbol{x}))$ & $F(M(W(\boldsymbol{x})))$ in Table 9.

## 5.2 Characteristics of Adversarial Signature

**Imperceptibility.** This is enforced by Eq. (4). The imperceptibility is demonstrated by Table 1-2, Figure 3 for the outputs of $W$; and Table 3-4, Figure 4 for the outputs of $\hat{G}$.

**Persistency. (1)** To make the signature in $\hat{\boldsymbol{y}}$ hard to be invalidated by image transformations, it is learned with data augmentation (see Section 4). According to Table 7, $F$ has gained robustness against the image transformations. **(2)** A possible adaptive attack from a malicious user may involve obtaining the inverse function of $W$, namely the restoration attack mentioned in Proposition A.5. To achieve this, $M$ learns to restore the original image $\boldsymbol{x}$ from the signed image $\hat{\boldsymbol{x}}$: $L_{\mathrm{M}} = \|M[W(\boldsymbol{x})] - \boldsymbol{x}\|^2$. Accordingly, the classifier $F$ has to recognize the outputs of $M$ by an extra loss term on top of Eq. (3): $L_{\mathrm{aux}} = \mathbb{E}_M\{\log(1 - F(M[W(\boldsymbol{x})]))\}$. In the implementation, we approximate the expectation over $M$ using multiple snapshots of $M$ jointly trained with $W, F$. The experimental results on FFHQ can be found in Table 9 and Fig. 6. The default setting (Table 1) is without the noise $\boldsymbol{e}$ (see Section 3.1), nor the $L_{\mathrm{aux}}$. When both the noise $\boldsymbol{e}$ and $L_{\mathrm{aux}}$ are applied, it is still difficult to remove the adversarial signatures even if the proposed method is disclosed to malicious users. The results support Proposition A.5.

**Inevitability.** Once the generative model is fine-tuned, the adversarial signature will be inevitably included in its outputs. Restoring $G$ from $\hat{G}$ may require access to the training images without signatures, with which a malicious user can already train new generators instead of using $\hat{G}$.

**Efficiency.** (1) Inference: Our method only changes the generative model parameters. The inference cost for $\hat{G}$ is identical to that of $G$. (2) Training: Assume $r$ generative models are to be released one by one. The cumulative complexity of re-training a detector every time upon the release of a new generator is $O(r^2)$. In contrast, the complexity of the proposed method is $O(r)$, because $W$ and $F$ are reused once trained. Our method is efficient in terms of complexity.

## 6 Conclusions

The proposed method aims to modify a given generative model, making its outputs more recognizable due to adversarial signatures. The adversarial signature can also carry additional information for tracking the source of generated images. The experimental results on two datasets demonstrate the effectiveness of the proposed method.

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

## A  APPENDIX

### A.1  ADDITIONAL RESULTS

Table 6 in the main paper shows the effect of varying the parameter $\lambda$ on the PSNR, FID and classification accuracy. Here we visualize the signed images with different $\lambda$ in Fig 7. We can see that the signed images are almost visually indistinguishable from the original images for $\lambda \in [1e-5, 0.1]$.

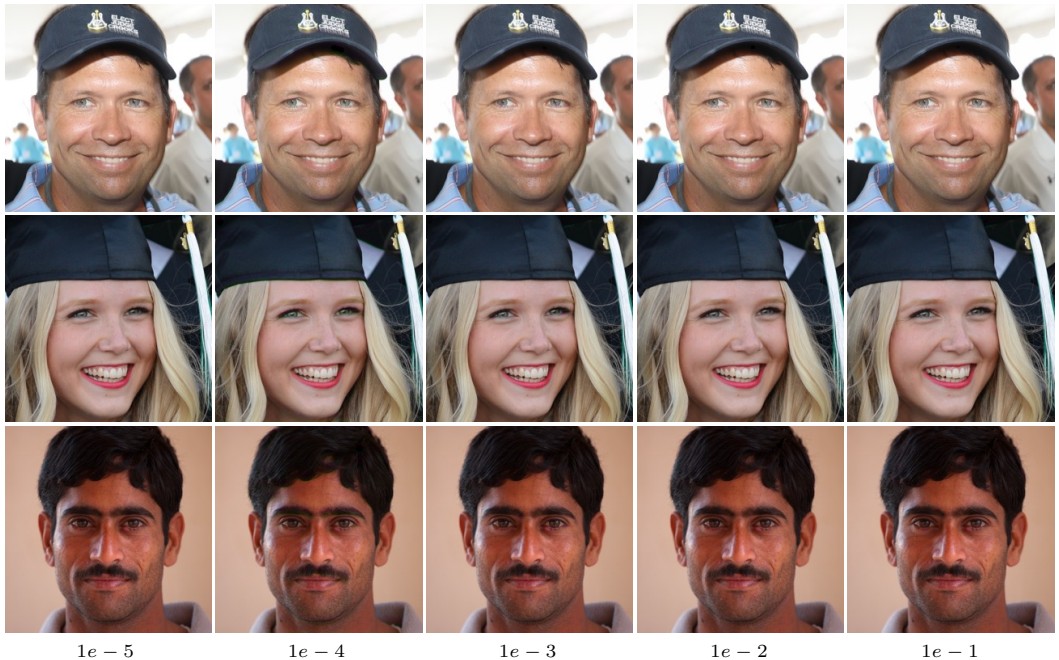

| $1e-5$ | $1e-4$ | $1e-3$ | $1e-2$ | $1e-1$ |

Figure 7: Visualization of the signed images with varying $\lambda$.

To further demonstrate the robustness of the proposed method, here we report the accuracy with JPEG comparison and center cropping in Table 10. The results with different image format and resolutiosn are reported in Table 11. The results show that the proposed method is also robust to JPEG comparison, center cropping, and resolution change.

Table 10: Generated image detection accuracy with JPEG noise and center cropping

| JPEG quality | 85 | 75 | 65 | 55 | Crop size | 220 | 200 | 180 |
|---|---|---|---|---|---|---|---|---|
| Accuracy | 99.5 | 99.9 | 97.7 | 99.9 | Accuracy | 99.9 | 99.6 | 88.0 |

Table 11: Generated image detection accuracy with different image format and resolutions.

| Format | PNG | JPG | Resolution | 256 | 512 | 768 |
|---|---|---|---|---|---|---|
| Accuracy | 100.0 | 99.9 | Accuracy | 100.0 | 100.0 | 100.0 |

Table 12 reports the comparisons of our method to the post-hoc watermarking method[2] used by StableDiffusion. As we have discussed in Sec. 3.1, our method applies optimal watermarks that

---

[2]https://github.com/ShieldMnt/invisible-watermark

introduce the smallest amount of perturbation to the original images and therefore have less negative impacts on the image quality. This can be seen from the higher PSNR of our method compared to traditional watermarks. Table 13 reports the comparison results to a watermarking method for diffusion models Zhao et al. (2023b). Our method achieves a higher detection accuracy and lower FID, which indicates that our method yields more accurate detection with better image quality.

Table 12: Comparisons to post-hoc watermarking approaches.

| Method | Post-hoc | Ours |
|--------|----------|------|
| PSNR   | 38.9     | 51.4 |

Table 13: Comparisons to watermarking of the diffusion models.

| Method | Zhao et al. (2023b) (4 bit) | Zhao et al. (2023b) (128 bit) | Ours |
|--------|------------------------------|-------------------------------|------|
| FID $\downarrow$ | 1.37 | 4.79 | 0.52 |
| Acc $\uparrow$ | 65.0 | 99.9 | 100.0 |

## A.2 PROOF OF PROPOSITIONS

**Proposition A.1.** *(**Imperceptibility**) There exist optimal pairs of signature injector $W$ and classifier $F$: $\mathbb{R}^n \mapsto \{0, 1\}$, so that for any image $\forall \boldsymbol{x} \in \mathbb{R}^n$, $\forall \epsilon > 0$, its distance to the signed image $W(\boldsymbol{x})$ is smaller than $\epsilon$, and $F$ correctly discriminates them,* i.e., $\|W(\boldsymbol{x}) - \boldsymbol{x}\| < \epsilon$, and $F(W(\boldsymbol{x})) \neq F(\boldsymbol{x})$.

*Proof.* For simplicity, we consider the case when $x \in \mathbb{R}^1$. Let $W(x)$ be an arbitrary irrational number within $(x - \epsilon, x + \epsilon)$ when $x$ is rational, and otherwise an arbitrary rational number within $(x - \epsilon, x + \epsilon)$. Let $F$ be a classifier that discriminates rational/irrational numbers. This pair of $W, F$ satisfies the given condition, and proves the existence of optimal watermarking systems.

*Remark* A.2. The $W, F$ presented in the proof are not feasible for implementation in practice. However, when $W, F$ are deep neural networks, the existence of adversarial samples Szegedy et al. (2013) implies that one can find a $W(\boldsymbol{x})$ that flips the prediction of $F$ while being very close to $\boldsymbol{x}$.

**Proposition A.3.** *Let $\boldsymbol{e}$ be a zero-mean, positive-variance additive noise. There exist noise augmented $W, F$ that satisfy the following condition: $\forall \epsilon > 0$, $\mathbb{E}_{\boldsymbol{e}}[\|W(\boldsymbol{x} + \boldsymbol{e}) - \boldsymbol{x}\|] < \epsilon$ and $F(W(\boldsymbol{x})) \neq F(\boldsymbol{x})$.*

*Proof.* We can prove the existence of such $W, F$ by constructing an example similar to the one in Proposition A.1 and set $\boldsymbol{e}$ to a rational noise. The existence of such $W, F$ can be proved in a similar way as Proposition A.1, by setting $\boldsymbol{e}$ to a rational noise. Then we have $\mathbb{E}[\|W(\boldsymbol{x} + \boldsymbol{e}) - \boldsymbol{x}\|] = \mathbb{E}[\|W(\boldsymbol{x} + \boldsymbol{e}) - (\boldsymbol{x} + \boldsymbol{e}) + \boldsymbol{e}\|] \leq \mathbb{E}[\|W(\boldsymbol{x} + \boldsymbol{e}) - (\boldsymbol{x} + \boldsymbol{e})\|] + \mathbb{E}[\|\boldsymbol{e}\|] < \epsilon$.

**Lemma A.4.** *Let $\boldsymbol{x}$ and $\boldsymbol{e}$ be zero-mean positive-variance random variables. For any non-constant mapping $M$, we have $\mathbb{E}_{\boldsymbol{x}, \boldsymbol{e}}[\|M(W(\boldsymbol{x} + \boldsymbol{e})) - \boldsymbol{x}\|] > 0$.*

*Proof.* Assume that $\mathbb{E}[\|M(W(\boldsymbol{x} + \boldsymbol{e})) - \boldsymbol{x}\|] = 0$. Then $\forall \boldsymbol{x}, \boldsymbol{e}$, $M(W(\boldsymbol{x} + \boldsymbol{e})) = \boldsymbol{x}$. If we let $\boldsymbol{x} = \boldsymbol{0}$, then $M(W(\boldsymbol{e})) = \boldsymbol{0}$, which is contradictory to the definition of $M$. Since the equal sign does not hold, and an L-2 norm is always greater than or equal to 0, we have $\mathbb{E}[\|M(W(\boldsymbol{x} + \boldsymbol{e})) - \boldsymbol{x}\|] > 0$.

**Proposition A.5.** *(**Persistency**) The noise augmented $W, F$ stated in Proposition A.3 is robust to image restoration attack, as optimizing $\min_M \mathbb{E}_{\boldsymbol{x}, \boldsymbol{e}}[\|M(W(\boldsymbol{x} + \boldsymbol{e})) - \boldsymbol{x}\|]$ will result in $M$ being an identity mapping.*

*Proof.* As shown in Proposition A.3, $\forall \epsilon > 0$, $\mathbb{E}[\|W(\boldsymbol{x} + \boldsymbol{e}) - \boldsymbol{x}\|] < \epsilon$. According to Lemma A.4, we have $\mathbb{E}[\|M(W(\boldsymbol{x} + \boldsymbol{e})) - \boldsymbol{x}\|] > 0$. Therefore, for any mapping $M$, $\mathbb{E}[\|W(\boldsymbol{x} + \boldsymbol{e}) - \boldsymbol{x}\|] \leq \mathbb{E}[\|M(W(\boldsymbol{x} + \boldsymbol{e})) - \boldsymbol{x}\|]$. Hence, $W(M(\boldsymbol{x})) = M(\boldsymbol{x})$ is the solution for $\min_M \mathbb{E}[\|M(W(\boldsymbol{x} + \boldsymbol{e})) - \boldsymbol{x}\|]$.

A.3   BROADER IMPACT

This work is intended to develop a system to mitigate the risk of image generation models by tracking the source of generated images based on signatures. Malicious users may attack this system with fake signatures, *e.g.* by adding a signature on a real image to make it classified as generated, and compromise the credibility of true information. Potential mitigation strategies include gated release of the watermark injectors, the use of longer multi-bit code and only releasing the code to the corresponding owners of generative models.

A.4   LIMITATIONS

**(1)** The binary code length $n$ limits the amount of additional information it can represent. However, most relevant works are merely designed in the $n = 1$ case.

**(2)** We assume the training dataset without adversarial signature is not available to malicious users. Once it is available, the malicious users may erase the signatures by fine-tuning the model on the original training set. Table 14 demonstrates the trend of a decaying detection accuracy of the signed LDM model, along the fine-tuning process on clean images. Nevertheless, its accuracy still remains higher than a random classifier after 2200 steps, which means the signatures can persist for a reasonable number of fine-tuning steps. Besides, preventing the signature decay when the attacker fine-tunes the model on clean images is very challenging and not yet explored in literature.

Table 14: Detection accuracy decay during the fine-tuning process of the generative model on original clean image training set.

| Fine-tuning steps | 400 | 800 | 1200 | 1400 | 1800 | 2200 |
|---|---|---|---|---|---|---|
| Accuracy | 100.0 | 99.8 | 98.5 | 88.3 | 74.9 | 64.8 |

While further fine-tuning on clean images implies the possibility to "erase" signatures, in practical scenarios, the attackers usually cannot access the original training data and sufficient computational resources. Those factors can effectively lower the practical risk of potential signature erasing by attackers. When both data and computing resources are available to attackers, they can already train an arbitrary clean generative model from scratch, in which case the persistence to fine-tuning will turn ineffective in preventing any malicious use.

**(3)** The proposed method requires finetuning a pre-trained generative model to embed the signature. A direction for future work is to explore the training-free framework to secure deep generative models, *e.g.* by directly modifying model parameters to further reduce the cost.

A.5   THREAT MODEL

Our threat model assumes the signed image generator is released to the public while the training datasets and the signature detector are kept private. Two use cases of our meethod are (1) reducing the risk of the abuse of generated images, and (2) protect the digital copyright and intellecgual property of the pre-trained generator.

A.6   DIFFERENCE FROM MODEL POISONING ATTACKS

There is a slight resemblance between the goal of poisoning attacks and the proposed signature. However, they are completely different in the problem setting. The poisoning attack aims to mix carefully designed data samples into the training data, and hence the poisoning attackers are releasing training data, instead of any pre-trained model. In contrast, the threat model of our method is that the model trainer only releases the pre-trained generator (with signature) to the public, while keeping the training data and the corresponding detector private. Thus, poisoning attacks are incompatible with our problem settings and threat model, despite the slight resemblance in their goals.

## A.7 HARDWARE

The signature injector is trained on an RTX A6000 GPU.

The generative models are finetuned using 4 RTX A6000 GPUs.

