# OpenReview forum: "Securing Deep Generative Models with Universal Adversarial Signature"
_ICLR.cc/2024/Conference — Submitted to ICLR 2024_

### Official Review · Reviewer_D7DW · 2023-10-29

**Soundness:** 2 fair
**Presentation:** 2 fair
**Contribution:** 2 fair
**Rating:** 3
**Confidence:** 3

**Summary:**

The paper delves into the topic of adversarial signatures within the realm of generative models. At its core, the research introduces a method that involves the integration of a signature injector and a classifier. The ideal scenario is when the classifier can distinguish images that have been signed from the original, clean images based on subtle, almost imperceptible alterations made by the injector.

The research also explores the idea of securing generative models by embedding adversarial signatures directly into the model's parameters, potentially through a fine-tuning process. By doing so, the outputs from these secured generators inherently carry the adversarial signatures, making them detectable by the classifier.

The methodology involves processing training data with the signature injector to produce signed images. An existing generative model is then fine-tuned using these signed images, ensuring that the images it generates carry the adversarial signatures.

**Strengths:**

Originality:
Innovative Approach: The paper introduces a unique method centered around adversarial signatures within the realm of generative models. This approach, which involves the integration of a signature injector with a classifier, offers a fresh perspective in the domain of adversarial defenses.

Quality:
Comprehensive Methodology: The paper's methodology, which encompasses the processing of training data with a signature injector followed by the fine-tuning of an existing generative model, is thorough. This structured approach suggests a well-thought-out experimental setup.


Clarity:
Structured Presentation: The paper appears to be well-organized, with clear sections detailing the introduction of the signature injector and classifier, the methodology, and the broader context of adversarial signatures within generative models. This structured approach aids in understanding the paper's flow and main contributions.

Significance:
Addressing a Crucial Challenge: The paper's focus on adversarial signatures within generative models addresses a significant challenge in deep learning. Given the importance of adversarial defenses in various applications, the paper's contributions in this area are relevant.

**Weaknesses:**

Originality:
Delineation from Existing Methods: The paper could benefit from a clearer differentiation from existing methods in the domain of adversarial signatures or defenses. Without specific comparisons or benchmarks, it's challenging to gauge the unique contributions of the proposed method.

Quality:
Lack of Detailed Experimental Results: The paper's methodology is described, but without detailed experimental results or comparisons, it's difficult to assess the effectiveness and robustness of the proposed approach. Comprehensive experiments or evaluations against benchmark methods would enhance the paper's credibility.

Potential Overfitting: The approach of fine-tuning a generative model with signed images raises concerns about overfitting. If the model is too closely tailored to the signed images, its generalizability to unseen data or different adversarial attacks might be limited.

Clarity:
Need for Enhanced Technical Details: While the paper presents its concepts, more in-depth technical explanations or visual aids, such as diagrams or flowcharts, would help readers better understand the intricacies of the proposed method.

Significance:
Unclear Practical Implications: The paper lacks a discussion on the practical implications or real-world applications of the proposed method. Understanding how this approach can be applied in real-world scenarios or its impact on existing systems would enhance its significance.
Questions on Scalability and Generalizability: The paper's focus on a specific approach to adversarial signatures within generative models raises questions about the method's scalability to larger datasets or more complex models. Additionally, its generalizability to other types of adversarial attacks or different domains remains unexplored.

In summary, while the paper presents valuable contributions, there are significant areas that need further exploration, validation, and clarification. Addressing these weaknesses could provide a clearer understanding of the paper's contributions and potential areas of improvement.

**Questions:**

Comparison with Existing Methods:
How does your approach to adversarial signatures differentiate from existing methods in the domain? Can you provide specific comparisons or benchmarks that highlight the advantages of your method?

Experimental Validation:
Could you provide more detailed results of your experiments, especially in comparison with state-of-the-art methods? How does your method perform in terms of robustness and effectiveness against various adversarial attacks?

Overfitting Concerns:
How do you address potential overfitting when fine-tuning the generative model with signed images? Have you conducted experiments to assess the model's generalizability to unseen data or different types of adversarial attacks?

Practical Implications:
What are the real-world applications of your proposed method? How does it impact existing systems or applications that utilize generative models?

Scalability and Generalizability:
How scalable is your method, especially when applied to larger datasets or more complex models? Have you tested its generalizability across different domains or types of adversarial attacks?

**Details Of Ethics Concerns:**

There were no immediate ethical concerns identified.

---

> ### Author Response · Authors · 2023-11-23
>
> ## The paper could benefit from a clearer differentiation from existing methods.
>
> The difference between our method and existing related work is described in the manuscript.
> (1). As described in paragraph 2 in the introduction section, as well as paragraph 3 in the related works section, post-hoc watermarking methods are decoupled from the generative model, which can be easily bypassed by a malicious user. Conversely, our method is to inject the proposed signature into a pre-trained generative model, making its generated images more detectable.
> (2). As described in paragraph 2 in the introduction section, as well as paragraph 2 in the related works section, the existing generated image detectors suffer from a performance drop against generators that are not seen during the training process. In contrast, our method does not suffer from the issue as long as the image is generated from a model with the proposed signature injected.
> (3). As described in paragraph 2 in the related works section, different from deep fake detectors that are specifically designed for human images, our proposed method is general-purpose generated image detection.
> (4) As described in paragraph 4 in the related works section, neural network fingerprinting methods have to be re-trained against newly appeared generators. In contrast, our detector can be reused for any future generator as long as the proposed signature is injected.
>
> ## Without specific comparisons or benchmarks, it's challenging to gauge the unique contributions of the proposed method.
>
> We presented detailed experiments in section 4.1 and section 4.2 to demonstrate the effectiveness of our proposed method. We compare our proposed method with the state-of-the-art methods in section 4.3. Our method outperforms the existing methods by a margin. Our unique contributions are summarized in the last paragraph of the introduction section.
>
> ## The paper's methodology is described, but without detailed experimental results or comparisons, it's difficult to assess the effectiveness and robustness of the proposed approach.
>
> We presented detailed experiments in section 4.1 and section 4.2 to demonstrate the effectiveness of our proposed method. We compare our proposed method with the state-of-the-art methods in section 4.3. We humbly ask the reviewer for a specific description of which experiment is missing from the manuscript, and we will add them to the manuscript.
>
> ## The approach of fine-tuning a generative model with signed images raises concerns about overfitting.
>
> We understand the reviewer’s concern. However, our experiments use the conventional dataset split, where the model outperforms existing methods on unseen images. This means our method has learned to generalize against images unseen during the training phase. All the experimental results demonstrate that our method does not suffer from overfitting issues.
>
> ## While the paper presents its concepts, more in-depth technical explanations or visual aids, such as diagrams or flowcharts, would help readers better understand the intricacies of the proposed method.
>
> We provided diagrams in Figure 1 to help readers better understand our proposed method. We provided in-depth technical descriptions of the proposed method in Section 3.
>
> ## The paper lacks a discussion on the practical implications or real-world applications of the proposed method.
>
> Thanks for the suggestion. Here are some examples of real-world applications. Our method can be adopted by model trainers who plan to publish pre-trained generative models but not their original training dataset. By adding signatures in a pre-trained image generative model, the model trainer can, for instance, protect intellectual property and copyright on the pre-trained model. For instance, assume a pre-trained model is sold to several customers, who are not allowed to re-sell the model as per the contract. In this case, different signatures or binary codes can be injected to the model sold to different customers, which helps the copyright holder to identify the customer who violated the contract for further legal actions.
>
> ## The paper's focus on a specific approach to adversarial signatures within generative models raises questions about the method's scalability to larger datasets or more complex models.
>
> We conduct experiments on the FFHQ and ImageNet datasets, and three state-of-the-art generative models including LDM, ADM, and StyleGAN2, which are commonly used large-scale datasets and models in the literature.
>
> ## Additionally, its generalizability to other types of adversarial attacks or different domains remains unexplored.
>
> We emphasize that the proposed method is for detecting generated images, which is a completely different topic compared to adversarial attacks and defenses.

---

### Official Review · Reviewer_3Y41 · 2023-11-03

**Soundness:** 2 fair
**Presentation:** 2 fair
**Contribution:** 3 good
**Rating:** 6
**Confidence:** 4

**Summary:**

This paper focuses on the detection methods for images generated by generative models and proposes to track the generated images by using adversarial signatures to make them more easily recognized by the designed detectors. Specifically, this paper constructs a signature injector W for learning to generate adversarial signatures and a classifier F for learning how to detect the images generated by W. Then, W and F are jointly trained. After that, the samples generated by W are then applied to finetune the original generative model G to obtain G'. The author elaborately designed the loss function as well as the binary code, and it has been shown through extensive experiments that the method can achieve good results.

**Strengths:**

(1)	This paper introduces adversarial examples into the detection of images generated by generative models, combined with joint training and watermark fine-tuning, which is novelty;
(2)	For the generation model, this paper investigates the latest diffusion-based generation model, which is of good practical significance under the common use of AIGC nowadays;
(3)	This paper is well-structured and logical. The author reviews the effectiveness and limitations of the proposed method from multiple perspectives. It points out the existing challenges, and gives possible solutions.

**Weaknesses:**

(1)	The two SOTA detection methods compared in the experiment are against the CNN-based and GAN-based generative model, whether there is any relevant paper for the watermarking of the diffusion model at present, if so, please supplement;
(2)	Missing a lot of experimental data on ImageNet, please add results comparing with SOTA on ImageNet;
(3)	Some formatting issues: (1) Please cite the graphs in order; (2) Please distinguish between periods and semicolons within the algorithm; (3) Please give the value of lamda in Figure 7.
(4)	The authors should indicate possible future directions in conclusion

**Questions:**

(1)	Fine-tuning an arbitrary G with samples generated by W to get G' is not unseen, in a sense it is inserting a backdoor that allows F to be better detected;
(2)	What is the difference between binary coding and multiclassification loss? What is the advantage of the binary coding?
(3)	I'm curious whether some of the latest adversarial defense methods can break this adversarial signature, such as diffusion-based purification. It would be great to add some experimental results in this scenario.
(4)	It mentioned in Sec5.2 that “The complexity of re-training a detector every time upon the release of a new generator is O(r^2)”, I wonder how to calculate this. For each new generator, the method of re-training a detector only needs to re-training once for the new generator.

---

> ### Author Response · Authors · 2023-11-23
>
> We thank the reviewer for the comments and suggestions. We have corrected the formatting issues in the revised paper and we answer the reviewer's questions as follows.
>
> # Compare to watermarking of the diffusion models
> We compare the performance of our method to a diffusion watermarking method [a] in the revised paper (also posted here). Our method achieves a higher detection accuracy and lower FID, which indicate that our method yields more accurate detection with better image quality.
>
> | Method | FID | Acc |
> | --- | --- | --- |
> | [a] 4 bit | 1.37 | 0.65 |
> | [a] 128 bit | 4.79 | 0.999 |
> | Ours | 0.52 | 1.00 |
>
> [a] Yunqing Zhao, Tianyu Pang, Chao Du, Xiao Yang, Ngai-Man Cheung, and Min Lin. A recipe for
> watermarking diffusion models. arXiv preprint arXiv:2303.10137, 2023b
>
> # Experimental data on ImageNet; future directions
> We included experimental results on ImageNet in Table 3 of the paper and possible future directions in Sec. A.4 (3) of the appendix.
>
> # Clarify regarding unseen generators and finetuning
> As we discussed in the paper (Sec 4 in the supplementary material), one limitation of the proposed method is the need for finetuning to embed signatures into a generative model. The training-free approach is an interesting direction for future work.
>
> # Difference between binary coding and multiclassification loss
> According to our observation, the performance difference between the binary classification loss and multi-class classification loss is negligible. We choose to use binary classification loss for every bit of the binary coding for simplicity.
> The advantage of binary coding is that it can carry more information (such as identity) than the 1-bit information about whether the image is generated or not. With the binary code, the owner of a specific generative model (with signature injected) will be able to inject the model identity information for protecting intellectual property.
>
> # Diffusion-based purification
> Diffusion-based purification can potentially remove any watermarks. However, it will impact the fidelity of the image, or even lose or change the original image contents during the process. Therefore, diffusion-based purification is not an ideal approach for removing watermarks/signatures.
>
> # $O(r^2)$ complexity of re-training a detector
> We are sorry for the clarity issue in our language. To be precise, this is the cumulative time complexity for the whole process of keeping the detector up-to-date. We will revise the manuscript to remove the ambiguity. The following is the elaboration on why the complexity is $O(r^2)$.
>
> * Upon the release of the first generative model, a generated image training set has to be collected for training the detector. We denote the time complexity in this case as 1.
> * Upon the release of the second generative model, the training dataset grows to two times larger than the previous step, because the training dataset must involve all the two models. In this case, the time complexity is 2.
> * Upon the release of the r-th generative model, the complexity for re-training is r.
> Thus, the cumulative complexity for keeping the detector up-to-date is $O(r(r-1)/2) = O(r^2)$.
> * We made this implicit in the manuscript by using the phrase “re-training a detector every time upon…”. We have made it more clear in the revised paper.
>
> In contrast, since we can always reuse the same signature injector $W$ and the corresponding detector $F$, the cumulative complexity of our method is merely $O(r)$, which only involves fine-tuning the generators for signature injection.

---

### Official Review · Reviewer_ce6G · 2023-11-05

**Soundness:** 3 good
**Presentation:** 3 good
**Contribution:** 3 good
**Rating:** 6
**Confidence:** 5

**Summary:**

The paper presents a simple idea of finetuning image generator models to embed an adversarial noise signature into the generated images, so that the resulting images can be easily detected by a classifier. This has been achieved in two steps: first learning a signature injector W together with the detector F, and then finetuning the generator G to produce these signatures. Experiments have shown that the inserted signatures are easily detectable by the detector F.

**Strengths:**

1) The papers deals with an important problem regarding the safety of generative models and proposes an innovative solution.
2) The proposed solution intuitively makes sense and is also feasible in practice.
2) The paper is well-written and easy to follow.

**Weaknesses:**

There are five main concerns about the proposed solution.

1) Firstly, the model F has been primarily described as a binary classifier whose goal is to distinguish between real images and "signed" synthetic images. What happens when the classifier F is fed with an "unsigned" synthetic image (i.e., an image generated from the original generator instead of the finetuned generator)? Shouldn't it be trained in such a way to detect "unsigned" synthetic images to the best possible extent along with "signed" synthetic images. Otherwise, why is a deep model such as ResNet-34 required to detect such "signed" images (a much smaller network should be enough to perform this relatively simpler task)?

2) The proposition 3.3 on persistency against image restoration attacks does not appear to be logical. Why would an attacker try to add noise e to the original image and learn a model M such that M(W(x+e)) = x. Instead the goal of the attacker would be to find e such that W(x)+e = x. In fact, even this image restoration task would not be required if the goal of the attacker is to simply bypass the detector F. The attacker has to just mount a simple adversarial attack to fool the detector F, i.e., add noise to W(x) such that F(W(x)) = real.

3) In terms of experiments, it is surprising that there is no comparison with a simple post-hoc watermarking approach (just add a unique generator-specific watermark to the generated image and categorize images with the watermark as synthetic), though this idea has been discussed in the introduction. It is true that the watermark will be decoupled from the generator, but it would still achieve the stated goals (imperceptibility, persistence, provenance of the generator, etc.). Also, the main paper does not talk about the ability to the detector to work under various image transformations. Only the appendix talks briefly about JPEG compression and cropping. How will the proposed detector work under various image transformations such as different levels of lossy compression, affine transformations, resolution and image format changes, etc.

4) Finally, there has to be more clarity on the threat model for the proposed solution. Specifically, who owns the generator and who will own the detector and what are their motivations? For instance, if the owner of the generator model is not honest (they want to hide their identity), why would they insert the signature in the first place? If a third-party is finetuning the generator, why would the users trust the finetuned model released by the third-party in place of the original model released by the owner. Similarly, who will own the detector and what happens when the owner of the detector is not honest? The proposed solution envisages that both the signature detector and detector are learned jointly. How will this be possible if the owners are different? Who will enforce the rule that every generated image must be signed? If there is no enforcer, the problem again boils down to detecting unsigned synthetic images where generalization is the core issue.

5) How is the proposed approach different from model poisoning attacks? The goal is to insert an "imperceptible" trigger pattern into the generated images such that the machine learning model will be easily triggered by these patterns and output  a specific decision (in this case, the synthetic class). One could potentially take any real vs. synthetic classification model and poison it to achieve this goal. Then, one has to just add the trigger to the generated images.

**Questions:**

Please see weaknesses.

---

> ### Author Response · Authors · 2023-11-23
>
> # Joint detection of "unsigned" synthetic images along with "signed"
> The joint detection of “unsigned” images with “signed” images is an interesting direction, which we will explore for future work. However, we would like to point out that detectors trained with “unsigned” images will not generalize well on unseen images, and meanwhile result in an $O(r^2)$ cumulative complexity for keeping the detector up-to-date, as explained in section 5.2. In contrast, since the training of our proposed signature injector W and the signature detector $F$ is independent of the generator, they can be reused for fine-tuning any image generators, and hence lead to the $O(r)$ cumulative complexity, as explained in section 5.2.
>
> # The purpose of $M(W(x+e))$ and why not add adversarial noise
> In the use cases of our method, the detector is not supposed to be released to the public so the white-box adversarial attack is not feasible. The transferability-based attacks are not feasible either because our threat model assumes the model owner only releases the pre-trained generator while keeping the training data and detector private. Even if black-box attacks can still be used, they will be much less effective and will be costly to the attacker to achieve a reasonable attack success rate. Therefore, the major threat comes from ``restoration attacks" where the malicious user attempts to remove the signature from the generated image with a signature removal model $M$. The input to $M$ is a signed image $W(x)$ and $e$ is only added during training to enhance robustness. The experimental results suggest that our method remains valid even if the attacker attempts to remove the signature.
>
> # Comparison with a simple post-hoc watermarking approach
> We would like to clarify that a simple post-hoc watermarking approach does not satisfy the persistence goal. This is because malicious users can easily avoid watermarking by simply removing the post-hoc watermarking process, e.g. by deleting the corresponding lines of code from the inference script. In contrast, the proposed method couples the watermarking process with image generation by changing the model weights, making it difficult for malicious users to bypass watermarking (signature). In addition, as we have discussed in Sec 3.1 of the paper, our method applies optimal watermarks that introduce the smallest amount of perturbation to the original images and therefore have less negative impacts on the image quality. This can be seen from the higher PSNR of our method compared to traditional watermarks as shown in the Table below. The comparison results with post-hoc methods are included in the revised paper.
> | Method    | Post-hoc | Ours |
> | -------- | ------- | ------- |
> | PSNR | 38.9 | 51.4 |
>
> # Results with lossy compression, different image formats, and resolutions
> We have included the results with common image transformations including affine transformation in Table 7 in the paper. The results with different levels of lossy compression can be found in Table 10 in the appendix. We report the additional results with different image formats and resolutions in the revised paper. The results demonstrate that our method is robust to image format and resolution change.
>
> # The threat model
> We apologize for the confusion. We have included the description of the threat model in the revised paper.  Our threat model assumes the signed image generator is released to the public while the training datasets and the signature detector are kept private. We address the reviewer’s concerns in detail as follows,
> * In practice, the developers of the foundation models are usually prestigious research groups, who are usually honest, and the threat often comes from the individual users. In the use cases of our method, the model developer is responsible for injecting the signatures by finetuning. This makes it non-trivial for individual users to avoid the signatures as they are already embedded into the model weights.
> * We assume a trustworthy third-party will own the detector, for example, the government or organizations who want to regulate the use of generative AI. This third-party will distribute the signature injectors to the model developers to sign the models.
>
> # Difference from model poisoning attacks
> This work and model poisoning attacks are completely different in the problem setting.
> The poisoning attack aims to mix carefully designed data samples into the training data, and hence the poisoning attackers are releasing training data, instead of any pre-trained model.
> In contrast, the threat model of our method is that the model trainer only releases the pre-trained generator (with signature) to the public, while keeping the training data and the corresponding detector private.
> Thus, poisoning attacks are incompatible with our problem settings and threat model, despite the slight resemblance in their goals. We have included the discussions in the appendix of the revised paper.

---

### Official Review · Reviewer_uR7H · 2023-11-10

**Soundness:** 3 good
**Presentation:** 3 good
**Contribution:** 2 fair
**Rating:** 5
**Confidence:** 4

**Summary:**

Injecting a universal adversarial signature into an arbitrary pre-trained generative model, in order to make its generated contents more detectable and traceable.

**Strengths:**

The motivation is explained clearly.
The paper is well-written.

**Weaknesses:**

The performance with or without the adversarial signature should be presented.
The term universal in used incorrectly since the signature depends on each image.

**Questions:**

Please see above.

---

> ### Author Response · Authors · 2023-11-23
>
> # Performance with or without the adversarial signature
>
> The second (baseline) and the last rows (ours) in Table 5 of the paper report the performance with and without the adversarial signature. It can be seen that without the adversarial signature, the detection accuracy drops significantly. In contrast, with the signature injected, the classifier can accurately discriminate images produced by unseen generators.
>
>
> # The term universal is used incorrectly
>
> We thank the reviewer for the suggestion, and we agree that the term “universal” is ambiguous without elaboration. We have removed the term “universal” from the title and added more clarifications in the paper. In particular, the proposed signature is “universal” because (1) a single signature injector W can be re-used for fine-tuning arbitrary given generators; (2) a single signature detector F can be re-used for all generators with the signature injected. Namely, the training process of W and F are independent of image generators. As a result, W and F can be re-used all the time.

---

### Meta-Review · Area_Chair_HcdS · 2023-12-07

**Metareview:**

The authors propose to inject a universal adversarial signature into an arbitrary pre-trained generative model, in order to make its generated contents more detectable and traceable.
Basically, the authors have responded to reviewers' comments but the reviewers did not actively participate in the rebuttal process.
When the AC looks at the Reviewer D7DW's comments, it seems that, as mentioned by the authors, the comments are too general (but have no concrete evidence).
As for the comments from Reviewer ce6G and Reviewer Reviewer 3Y41, they are indeed constructive.
However, the authors fail to provide complete results in responding some comments. For example, the impact of diffusion-based purification on the proposed method needs to be studied.
The authors did not provide any evidence in this aspect.
Based on the above concerns, the paper is suggested to be rejected.

**Justification For Why Not Higher Score:**

Some comments were not responded with sufficient evidences.

**Justification For Why Not Lower Score:**

The authors have responded to some comments and this paper indeed contains some contributions.

---

### Decision · Program_Chairs · 2024-01-16

Reject